# Survival and clinicopathological characteristics of different histological grades of oral cavity squamous cell carcinoma: A single-center retrospective study

**Nan-Chin Lin[1,2], Jui-Ting Hsu[1], Kuo-Yang Tsai [ID][2,3]***

**1** School of Medicine, China Medical University, Taichung, Taiwan, ROC, **2** Department of Oral and Maxillofacial Surgery, Changhua Christian Hospital, Taiwan, ROC, **3** College of Nursing and Health Science, Da-Yeh University, Changhua, Taiwan, ROC

* 72837@cch.org.tw

**Data Availability Statement:** All relevant data are within the paper and its Supporting Information files.

## Abstract

The TNM staging system for oral squamous cell carcinoma (OSCC) provides clinicians a dependable foundation for patient prognosis and management decisions, but in clinical practice, treatment outcomes of patients with OSCC are sometimes unsatisfactory. This retrospective study investigated the association between survival and clinicopathological characteristics and histological grades of 2535 patients with OSCC. Additionally, the present study aimed to compare the predictive abilities of histological grades with other common prognostic factors. The enrolled patients were divided into three groups by two experienced pathologists into well-differentiated, moderately differentiated, and poorly differentiated groups, according to the WHO classification. Finally, we designed an observational, retrospective study based on the histological grading of tumors to compare their clinicopathological characteristics and conducted survival analysis among the three groups. Advanced tumor stage was diagnosed in 23.9%, 44.0%, and 55.1% of patients with grades 1–3 OSCC, respectively. By T status, T3 or T4 tumors were found in approximately 22%, 34%, and 40% of patients with grades 1–3 OSCC, respectively. By N status, lymph node metastases were found in 6.1%, 29.3%, and 45.9% of patients with grades 1–3 OSCC, respectively. Thus, significant survival differences were observed based on different OSCC histological grades. Meanwhile, in the multivariate (adjusted) analysis, N1 and N2 stages, extranodal spread, and poor differentiation were associated with a higher recurrence risk than the other common prognostic factors. In conclusion, 5% of patients in our study presented with poorly differentiated OSCC at diagnosis. Furthermore, grade 3 OSCC has worse prognosis and is more aggressive than grades 1 and 2 OSCC. In the future, we should focus on modifying individual therapy for poorly differentiated OSCC to achieve improved outcomes.

**Funding:** The authors received no specific funding for this work.

**Competing interests:** The authors have declared that no competing interests exist.

## Introduction

The TNM staging system of oral squamous cell carcinoma (OSCC) has historically provided clinicians a dependable foundation for patient prognosis and management decisions. Generally, the relative risk of neck lymph node metastasis of T1 and T2 tumors is 10% and 30%, respectively, whereas that of T3 and T4 tumors is distinctly higher [1, 2]. This is significant because the most important prognostic factor in OSCC is the presence of cervical lymph node metastasis, which results in the reduction of the overall survival of these patients by 50% [3, 4].

However, in clinical practice, the treatment outcome of OSCC is sometimes not satisfactory. Several studies have shown that even an early-stage tumor may cause a fatal outcome [5–7]. New parameters such as depth of invasion was included in the 8th edition of the American Joint Committee on Cancer's (AJCC) Cancer Staging Manual to improve their predictive value and different stage stratification and to explain patients previously considered to have early-stage tumor but with a poor survival.

The pathological parameters including pathological stage, histopathological grading as per the World Health Organization (WHO) [8], presence of vascular and perineural invasion, extracapsular spread, and positive surgical margins have been used as tumor prognostic factors in OSCC. In this situation, histological grade can be an important prognostic factor for treatment outcome. Evaluation of histological characteristics of a tumor plays an important role in the diagnosis of ablated tumor specimens, and efforts have been undertaken to predict clinical outcomes from histological findings.

In the present retrospective study, we sought to address the association between clinicopathological characteristics and survival of OSCC with different histological grades. We also aimed to compare the predictive abilities of histological grade to other common prognostic factors. To this end, we performed survival analysis based on different histological grading of tumors.

## Patients and methods

### Patients

This retrospective cohort study was approved by the Institutional Review Board (IRB) and Ethical Committee of Changhua Christian Hospital on April 22, 2020 (IRB number: 200312). All clinical data were obtained through chart review and the cancer registry center. In total, we identified 3248 patients who were diagnosed with OSCC and received treatment and follow-up at Changhua Christian Hospital between January 1, 2008 and December 31, 2018. The follow-up duration was from indexing to June 30, 2019. Exclusion criteria included patients who did not receive treatment per the AJCC cancer treatment guidelines (n = 41), whose anatomic site of the lip did not involve the mucosa (n = 121), who were lost to follow-up or had incomplete data (n = 140), who were initially diagnosed with recurrence or distant metastasis (n = 70), and who had not receive treatment at Changhua Christian Hospital (n = 341). Finally, 2535 patients were included and subsequently analyzed.

### Histopathological evaluation

Final pathological reports of resected tumor specimens from all 2535 patients were further investigated. Hematoxylin and eosin-stained slides were reviewed and independently graded by two experienced pathologists under a microscope; if they could not reach a consensus, the intervention of a third pathologist was engaged, and a final agreement was obtained for histopathological features according to the 2017 The College of American Pathologists oral cavity cancer guidelines [9]. Immunohistochemical confirmation for cytokeratin was performed to

detect poorly differentiated OSCC because features of squamous differentiation were minimal or absent. The enrolled patients were divided into three groups according to the degree of keratinization, nuclear pleomorphism, and mitosis rate based on the current WHO classification criteria of tumors of the head and neck: well differentiated (grade 1, n = 435), moderately differentiated (grade 2, n = 1964), and poorly differentiated (grade 3, n = 136) [8]. The WHO grading system, revised on the basis of Broders' classification [10] in 2005, involved microscope based searches for differences between tumor tissue and normal epithelium because of the lack of cellular differentiation. This classification was used routinely in biopsy and surgical specimen analyses [8]. The characteristics of the three grades are shown in Fig 1.

## Clinicopathological parameters

The characteristics of the three groups that were analyzed included age at OSCC diagnosis, survival time, sex, pathological AJCC anatomic site, AJCC TNM stage (7th and 8th edition), recurrence, close margin (safe margin ≦ 1 mm), and extranodal spread as well as behaviors, including smoking, chewing betel nuts, and alcohol consumption. The anatomic site were subclassified into the alveolar ridge, anterior two-thirds of the tongue, buccal mucosa, hard palate, floor of the mouth, retromolar trigone, and lip mainly involving the mucosa. Death information was retrieved from the cancer registry center of Changhua Christian Hospital as well as the data renewed annually by the Health Bureau of Changhua city. Finally, we designed an

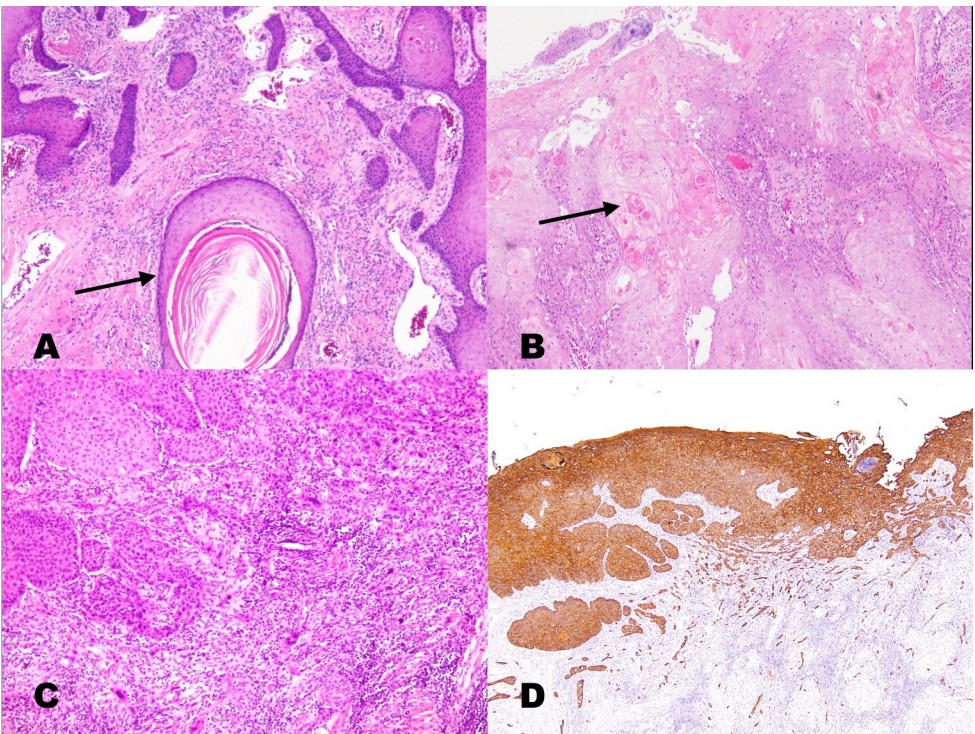

**Fig 1.** a. A keratinizing (well-differentiated) squamous cell carcinoma showing large keratin pearls (arrows) and well-defined tumor cell outlines with slender cytoplasmic connections (intercellular bridges). Nuclear pleomorphism was mild, and the mitotic rate was low. b. Moderately differentiated OSCC (grade 2) presenting nests of basaloid tumor cells with peripheral palisading at the stromal interface. Intercellular bridges may have been present, but keratinization was focal (arrow). c. Poorly differentiated OSCC (grade 3) presenting confluent nests of tumor cells lacking keratinization or intercellular bridges. There is a slight background lymphocytic infiltrate, likely representing a host response. d. The identification of this grade 3 tumor as squamous carcinoma may require confirmatory immunostaining for cytokeratin, p63, or p40 (the same patient in Fig 1C with immunostaining for cytokeratin).

observational, retrospective study based on the histological grading of tumors to compare their clinicopathological characteristics as well as survival analysis among the three groups.

## Statistical analysis

Continuous variables are presented as means ± standard deviation, and categorical variables are presented as percentages. One-way analysis of variance was used to compare the continuous variables among the three groups. Chi-square test was used to compare the differences in the categorical variables among the three groups. The effects of clinicopathological factors on recurrence among patients with OSCC were examined using univariate and multivariate Cox proportional hazard models. Hazard ratios (HRs) and confidence intervals (CIs) were subsequently calculated. Estimates of the overall survival rates (OS) were calculated using Kaplan–Meier analyses. Comparisons of the group survival functions were performed using log rank tests based on OS. A p-value of <0.05 was considered statistically significant. All statistical analyses were performed using statistical package SPSS for Windows (version 16, SPSS, Chicago, IL, USA).

## Results

Our retrospective study enrolled 2535 patients who were divided into three groups: well differentiated (grade 1; group 1), moderately differentiated (grade 2; group 2), and poorly differentiated (grade 3; group 3). The results of our histology grading showed 435, 1964, and 136 patients with grade 1 (17%), grade 2 (78%), and grade 3 (5%) OSCC, respectively.

Fig 2A shows that the mean age at diagnosis was 59.0, 57.0, and 54.9 years for grade 1, grade 2, and grade 3 OSCC, respectively. Fig 2B shows that the mean survival time over 120 months (January 1, 2008–June 30, 2019) was 52.4, 43.9, and 33.1 months for grade 1, grade 2, and grade 3 OSCC, respectively. There was statistically significant difference in the mean age at diagnosis of cases and survival time (p < 0.001).

The clinicopathological characteristics are listed in Tables 1 and 2. Table 1 summarizes sex, stratified data of age at diagnosis, anatomic site of the tumor and the associations among the grade of tumor cell differentiation and habits of lifestyle, including smoking, betel nut

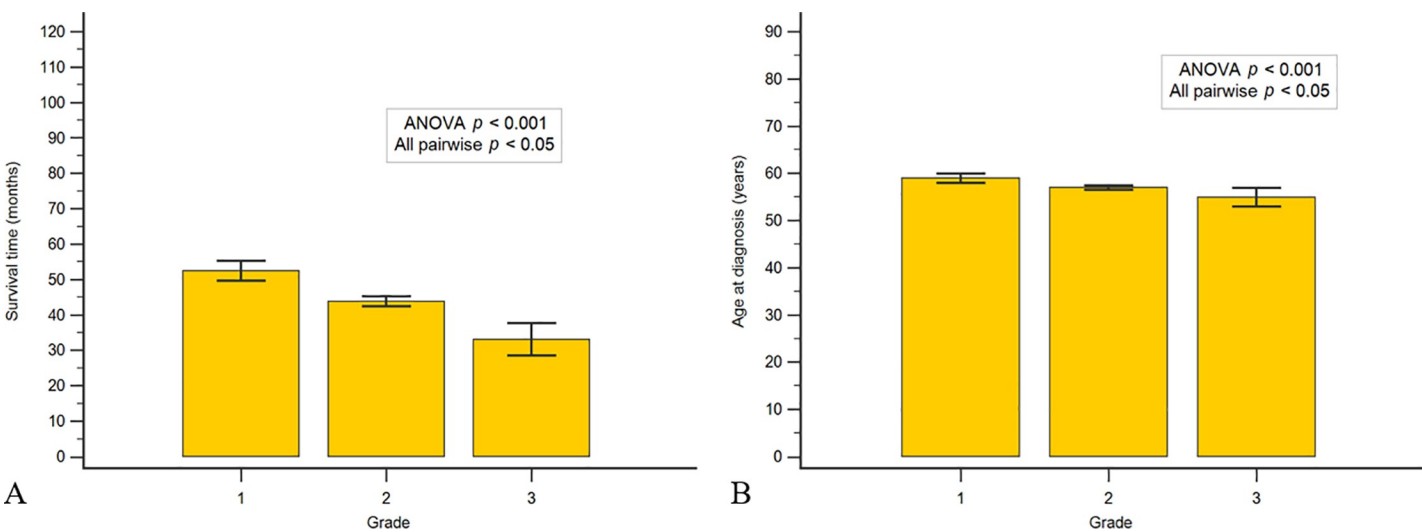

**Fig 2.** a. Mean age at diagnosis of 59.0, 57.0, and 54.9 years with grade 1 (n = 435), grade 2 (n = 1964), and grade 3 (n = 136) OSCC, respectively. b. Mean survival time over 120 months (from OSCC diagnosis to death or to June 30, 2019) of 52.4, 43.9, and 33.1 months with grade 1, grade 2, and grade 3 OSCC, respectively.

**Table 1. Summarized, stratified data of age at diagnosis, anatomic tumor site, and sex and associations among the grade of tumor cell differentiation and habits of smoking, betel nut chewing, and alcohol consumption.**

| | | Grade | | | | | | | | p-value |
| | | Grade 1 (n = 435) | | Grade 2 (n = 1964) | | Grade 3 (n = 136) | | Total (n = 2535) | | |
| | | N | % | N | % | N | % | N | % | |
|---|---|---|---|---|---|---|---|---|---|---|
| Age at diagnosis | ≤40 | 15 | 3.4 | 137 | 7.0 | 11 | 8.1 | 163 | 6.4 | <0.001 |
| | 41–50 | 75 | 17.2 | 434 | 22.1 | 37 | 27.2 | 546 | 21.5 | |
| | 51–60 | 156 | 35.9 | 701 | 35.7 | 48 | 35.3 | 905 | 35.7 | |
| | 61–70 | 127 | 29.2 | 448 | 22.8 | 20 | 14.7 | 595 | 23.5 | |
| | ≧71 | 62 | 14.3 | 244 | 12.4 | 20 | 14.7 | 326 | 12.9 | |
| Site of tumors | Alveolar ridge | 71 | 16.3 | 328 | 16.7 | 16 | 11.8 | 415 | 16.4 | <0.001 |
| | Anterior tongue | 82 | 18.9 | 587 | 29.9 | 52 | 38.2 | 721 | 28.4 | |
| | Buccal mucosa | 193 | 44.4 | 659 | 33.6 | 39 | 28.7 | 891 | 35.1 | |
| | Hard palate | 16 | 3.7 | 53 | 2.7 | 4 | 2.9 | 73 | 2.9 | |
| | Floor of mouth | 3 | 0.7 | 67 | 3.4 | 7 | 5.1 | 77 | 3.0 | |
| | Retromolar trigone | 15 | 3.4 | 110 | 5.6 | 7 | 5.1 | 132 | 5.2 | |
| | Mucosa of the Lips | 55 | 12.6 | 160 | 8.1 | 11 | 8.1 | 226 | 8.9 | |
| Gender | Female | 12 | 2.8 | 98 | 5.0 | 9 | 6.6 | 119 | 4.7 | 0.076 |
| | Male | 423 | 97.2 | 1866 | 95.0 | 127 | 93.4 | 2416 | 95.3 | |
| Smoking | No | 97 | 26.6 | 418 | 25.8 | 28 | 23.9 | 543 | 25.8 | 0.839 |
| | Yes | 267 | 73.4 | 1204 | 74.2 | 89 | 76.1 | 1560 | 74.2 | |
| | Unknown | | | | | | | 432 | | |
| Betel nut | No | 171 | 47.0 | 687 | 42.4 | 48 | 41.0 | 906 | 43.1 | 0.246 |
| | Yes | 193 | 53.0 | 935 | 57.6 | 69 | 59.0 | 1197 | 56.9 | |
| | Unknown | | | | | | | 432 | | |
| Alcohol | No | 183 | 56.0 | 714 | 49.7 | 47 | 46.1 | 944 | 50.6 | 0.079 |
| | Yes | 144 | 44.0 | 723 | 50.3 | 55 | 53.9 | 922 | 49.4 | |
| | Unknown | | | | | | | 669 | | |

p-value by chi-square test. Regarding sex, p trend = 0.024 In the alcohol group, p trend = 0.027.

Grade 1: well differentiated; Grade 2: moderately differentiated; Grade 3: poorly differentiated.

In the smoking and betel nut groups, 432 cases were excluded due to incomplete data.

In the alcohol group, 669 cases were excluded due to incomplete data.

chewing, and alcohol consumption. There was no significant sex difference among the three groups (p = 0.076). Comparing among three groups, the proportion of females was slightly increased with an increasing trend in grade of tumor cell differentiation (p trend = 0.024). Age at diagnosis was divided into five groups, and there were significant differences among the three groups (p < 0.001). In grades 1 and 2 OSCC, age at diagnosis was mainly in group 3 (51–60 years), followed by group 4 (61–70 years), and in grade 3 OSCC, age at diagnosis was mainly in group 3 (51–60 years), followed by group 2 (41–50 years). There were significant differences in the anatomic site of the tumor among the three groups (p < 0.001). Most tumors of grades 1 and 2 were located on the buccal mucosa (44.4% and 33.6%, respectively), and grade 3 tumors were mostly located on the anterior two-thirds of the tongue (38.2%). Regarding life-style habits, Table 1 shows no significant association between the grade of tumor cell differentiation and habits of smoking, betel nut chewing, and alcohol consumption. Nevertheless, with increasing tumor grade, we noted an increase in the proportion of drinkers (p = 0.027).

Table 2 shows a significant association between the grade of tumor cell differentiation and pathological stage (overall stage, T stage, and N stage), extranodal spread, and recurrence

**Table 2. Associations among the grade of tumor cell differentiation and tumor pathological stage, including overall stage, T stage, N stage, recurrence status, and extranodal spread.**

| | | Grade | | | | | | | | p-value |
|---|---|---|---|---|---|---|---|---|---|---|
| | | Grade 1 (n = 435) | | Grade 2 (n = 1964) | | Grade 3 (n = 136) | | Total (n = 2535) | | |
| | | N | % | N | % | N | % | N | % | |
| Stage | 1 | 249 | 59.1 | 712 | 36.8 | 27 | 20.5 | 988 | 39.7 | <0.001 |
| | 2 | 70 | 16.6 | 359 | 18.6 | 31 | 23.5 | 460 | 18.5 | |
| | 3 | 30 | 7.1 | 164 | 8.5 | 21 | 15.9 | 215 | 8.6 | |
| | 4 | 72 | 17.1 | 699 | 36.1 | 53 | 40.2 | 824 | 33.1 | |
| | Missing data | | | | | | | 48 | | |
| T stage | 1 | 262 | 60.2 | 812 | 41.5 | 36 | 26.5 | 1110 | 43.9 | <0.001 |
| | 2 | 77 | 17.7 | 477 | 24.4 | 46 | 33.8 | 600 | 23.7 | |
| | 3 | 25 | 5.7 | 109 | 5.6 | 17 | 12.5 | 151 | 6.0 | |
| | 4 | 71 | 16.3 | 560 | 28.6 | 37 | 27.2 | 668 | 26.4 | |
| | No residual tumor | | | | | | | 6 | | |
| N stage | 0 | 261 | 93.9 | 1096 | 70.7 | 59 | 54.1 | 1416 | 73.1 | <0.001 |
| | 1 | 10 | 3.6 | 154 | 9.9 | 18 | 16.5 | 182 | 9.4 | |
| | 2 | 7 | 2.5 | 273 | 17.6 | 29 | 26.6 | 309 | 16.0 | |
| | 3 | 0 | 0.0 | 27 | 1.7 | 3 | 2.8 | 30 | 1.5 | |
| | Incomplete data | | | | | | | 598 | | |
| N stage | Negative | 261 | 93.9 | 1096 | 70.7 | 59 | 54.1 | 1416 | 73.1 | <0.001 |
| | Positive | 17 | 6.1 | 454 | 29.3 | 50 | 45.9 | 521 | 26.9 | |
| | Incomplete data | | | | | | | 598 | | |
| Extra-nodal spread | No | 271 | 97.5 | 1318 | 85.0 | 80 | 73.4 | 1669 | 86.2 | <0.001 |
| | Yes | 7 | 2.5 | 232 | 15.0 | 29 | 26.6 | 268 | 13.8 | |
| | Incomplete data | | | | | | | 598 | | |
| Recurrence | No | 375 | 86.2 | 1563 | 79.4 | 94 | 69.1 | 2028 | 80.0 | <0.001 |
| | Yes | 60 | 13.8 | 405 | 20.6 | 42 | 30.9 | 507 | 20.0 | |

p-value by chi-square test. In the overall stage, N positive or negative group and extra-nodal spread group, p trend < 0.001.

Grade 1: well differentiated; Grade 2: moderately differentiated; Grade 3: poorly differentiated.

In the T stage group, 6 cases received curative chemoradiotherapy first, and the final pathological report showed no residual tumor.

In the overall stage, 48 cases were excluded due to poor data quality.

In the N stage and extranodal spread group, 550 cases were excluded due to operation without neck dissection and 48 cases were excluded due to poor data quality.

(p < 0.001). More than half (56.1%) of the patients with grade 3 OSCC but only 24.2% and 44.6% patients with grades 1 and 2 OSCC, respectively, were diagnosed with advanced tumor stage. By T status, T3 or T4 tumors were found in approximately 22%, 34%, and 40% of patients with grades 1–3 OSCC, respectively. Out of the patients with grade 3 OSCC, 45.9% were diagnosed with lymph node metastasis (N status positive), whereas only 6.1% with grade 1 and 29.3% with grade 2 were N status positive. Simultaneously, pathological N2 and N3 status were found in 29.4% of patients with grade 3 OSCC.

Table 2 also shows that there was a significant difference between recurrence status and extranodal spread among the three study groups (p < 0.001). Recurrence was recorded in 13.8%, 20.6%, and 30.9% of patients with grades 1–3 OSCC, respectively. Extranodal spread was recorded in 26.9%, 15.5%, and only 2.6% of patients with grade 3, grade 2, and grade 1, respectively.

Comparison of predictive power between pathologic grade and other common prognostic factors are shown in Table 3. In the multivariate (adjusted) analysis, during N1 and N2 stage,

**Table 3. The effect of clinicopathological factors on recurrence and non-adjusted and adjusted hazard ratio among OSCC patients.**

| Cox proportional-hazards regression analysis of recurrence | | | Recurrence | | Univariate analysis (crude) | | | Multiple analysis (adjusted) | | | | |
|---|---|---|---|---|---|---|---|---|---|---|---|---|
| | | Total | N | % | Hazard ratio | 95% CI | P-value | Hazard ratio | 95% CI | | | P-value |
| Stage | 1 | 988 | 160 | 16.1 | 1.000 | | | | | | | |
| | 2 | 460 | 88 | 19.1 | 1.235 | 0.952–1.603 | 0.111 | | | | | |
| | 3 | 215 | 35 | 16.2 | 1.085 | 0.752–1.564 | 0.663 | | | | | |
| | 4 | 824 | 220 | 26.7 | 1.954 | 1.593–2.396 | <0.001 | | | | | |
| T stage | 1 | 1110 | 188 | 16.9 | 1.000 | 0.158–2.566 | 0.526 | | | | | |
| | 2 | 600 | 126 | 21.0 | 1.322 | 0.208–3.405 | 0.810 | | | | | |
| | 3 | 151 | 28 | 18.5 | 1.301 | 0.198–3.481 | 0.798 | | | | | |
| | 4 | 668 | 170 | 25.4 | 1.749 | 0.276–4.489 | 0.880 | | | | | |
| N stage | 0 | 1416 | 209 | 14.8 | 1.000 | | | | | | | |
| | 1 | 182 | 46 | 25.3 | 1.851 | 1.345–2.547 | <0.001 | 1.523 | 1.019 | - | 2.276 | 0.040 |
| | 2 | 309 | 111 | 35.9 | 3.174 | 2.520–3.999 | <0.001 | 2.426 | 1.629 | - | 3.613 | <0.001 |
| | 3 | 30 | 10 | 33.3 | 5.259 | 2.777–9.962 | <0.001 | 3.140 | 0.939 | - | 10.500 | 0.063 |
| Extranodal spread | No | 1669 | 266 | 15.9 | 1.000 | | | 1.000 | | | | |
| | Yes | 268 | 102 | 38.0 | 3.156 | 2.509–3.970 | <0.001 | 1.666 | 1.129 | - | 2.458 | 0.010 |
| Close margin | No | 2455 | 491 | 20.0 | 1.000 | | | | | | | |
| | Yes | 80 | 23 | 28.8 | 1.527 | 1.005–2.320 | 0.047 | | | | | |
| Grade | Well | 435 | 60 | 13.8 | 1.000 | | | 1.000 | | | | |
| | Moderately | 1964 | 405 | 20.6 | 1.672 | 1.275–2.194 | <0.001 | 1.167 | 0.793 | - | 1.715 | 0.433 |
| | Poor | 136 | 42 | 30.9 | 3.029 | 2.041–4.495 | <0.001 | 1.973 | 1.167 | - | 3.336 | 0.011 |

Following period was from time at OSCC diagnosed to June 30, 2019.

extranodal spread and poor differentiation were observed to be associated with a higher recurrence risk than the other common prognostic factors listed in Table 3 (aHR = 1.5, 2.4, 1.7, and 2.0, respectively).

Kaplan–Meier curves for the three different histological grade tumors based on overall stage, early stages (stages I and II), advanced stages (stages III and IV), early T stages (T1 and T2), and without neck metastasis (pN0) are presented in Fig 3. These results show significant

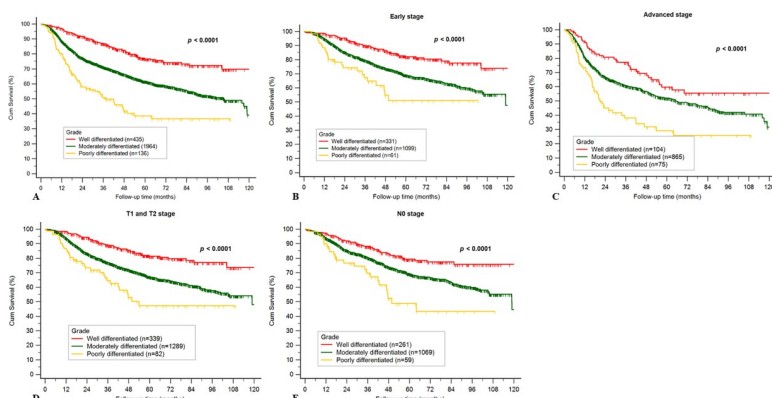

**Fig 3.** a. Kaplan–Meier curve for different grades of tumor cell differentiation in the overall stage. b. Kaplan–Meier curve for different grades of tumor cell differentiation at early stage. c. Kaplan–Meier curve for different grades of tumor cell differentiation at advanced stage. d. Kaplan–Meier curve for different grades of tumor cell differentiation with early T stage (T1 and T2). e. Kaplan–Meier curve for different grades of tumor cell differentiation at N0 stage.

differences in overall stage, early stage, advanced stage, early T stage, and N0 stage ($p < 0.001$). Based on the Kaplan–Meier analysis, a better survival condition was observed in patients with well-differentiated OSCC than in those with moderately differentiated or poorly differentiated OSCC.

## Discussion

Despite improved treatment modalities, clinicopathological prognostic factors remain insufficient for predicting recurrence and survival of OSCC; hence, the survival of patients has remained unchanged over the last few decades [11]. OSCC is known to present with verifiable histological behavior patterns. In our study, when compared with grades 1 and 2 OSCC, grade 3 OSCC was found in younger patients and was mostly located on the anterior two-thirds of the tongue. Patients with OSCC diagnosed as grade 3 tumors showed poor survival condition and higher risk of recurrence than other groups. Besides, histological grade of OSCC was observed to be a stronger predictive factor for predicting recurrence when compared with the other common prognostic factors.

In the present study, we found that the presence of high-grade histology was strongly associated with advanced T stage, neck lymph node metastasis, and extranodal spread, which subsequently contributed to advanced stage and prevalent recurrence. Similar conclusions have been deduced in previous research [12–18]. Kademani et al. presented 10-year survival among three different histological grade OSCCs in 233 patients, and their results were similar to those of our study [12]. However, our study compares survival among not only three grades of tumors but also with respect to different clinicopathological parameters. Larsen et al. showed that T1 and T2 tumors had better local control than that of T3 and T4 tumors; however, T4 tumors also had better local control than that of T3 tumors [13]. These results may have been related to sample quantity (total sample size, n = 142). Contrarily, we studied a larger sample size and emphasized the distinct survival differences among the three groups of early stage, early T stage, and N0. Niu et al. showed that T stage, N stage, and histopathological grade of tumor were significantly associated with recurrence among 168 patients in a North Chinese population [16]. In the present study, histopathological grade of tumor, as opposed to T stage, was a more powerful prognostic factor to predict recurrence. The same research team, Xu et al., reported similar survival conditions among three different histological grade OSCCs in 2036 patients, and also reported that histopathological grade was associated with many clinicopathological features [16]. Simultaneously, they revealed that histopathological grade was an independent prognostic factor for patients with early-stage OSCC but not for patients with advanced-stage OSCC [17]. However, in our study, histopathological grade was associated with not only the survival condition of patients with early-stage OSCC but also of those with advanced stage OSCC.

Importantly, we observed that grade 3 OSCC may be important in early-stage cases for indicating potentially poor treatment outcomes because survival of patients with early-stage tumor and grade 3 tumor was distinctly poorer than that of patients with grades 1 and 2 tumor. The commonly accepted TNM staging for cancer is currently used by clinicians to guide their management decisions, but treatment outcomes are sometimes unsatisfactory for patients with early-stage tumor. In clinical practice, histopathological grade of OSCC is not regarded as a prognostic factor in National Comprehensive Cancer Network (NCCN) guideline; however, histological grade may be used as a supplementary material to clinical practice and provide additional information for decision making.

Several studies have indicated that patients with grade 3 OSCC are more likely to present with neck lymph node metastasis and be associated with decreased survival as compared to the

patients with other tumor grades [12–19]. However, the sample sizes of these studies were not as large as that in our study. Moreover, to the best of our knowledge, the current study is the first to describe the association among age, anatomic tumor site, alcohol consumption, and tumor grade in Taiwan. In clinical practice, relatively young patients with early TNM OSCC stage, especially in the tongue, should be more intensively followed up if they have higher differentiation degree of tumors because this may lead to occult metastasis, extranodal spread, and poor survival.

Regarding prognostic factors, we found that N stage, extranodal spread, and grade 3 tumors were strongly associated with recurrence. No significant correlation was found between recurrence and overall TNM and T stage, and there was a surprisingly close margin in our study. This finding may be explained by the advances in radiotherapy. In clinical practice, when facing close margins in resected tumors, clinicians can further consider wider re-excision or radiotherapy to decrease recurrence. Perhaps, this could explain that the role of close margin was not as important as other prognostic factors.

There are several limitations of our study. First, in this study, data were collected from a single medical center in Taiwan. Oral cavity cancer is highly associated with betel nuts, popular in Taiwan, which leads to buccal mucosa cancer, indicating a predilection at our site that may differ from other geographical regions. According to data from the Health Promotion Administration, Ministry of Health and Welfare in Taiwan, oral cavity cancer mostly occurs in the buccal mucosa followed by the tongue [20]. Second, the gross and histologic features of squamous cell carcinoma involving the head and neck are similar to those of squamous cell carcinoma in other organs. However, the histological grading of invasive squamous cell carcinoma remains controversial and there is no general agreement on the best system or scoring scheme [21, 22]. Finally, in the present study, although pathological slides were graded by two experienced pathologists, at best, grading attempts were subjective, and not absolutely reproducible. However, the modifiers "well," "moderately," and "poorly" differentiated carcinoma are used by many pathologists. "Moderately differentiated" is a commonly utilized fallback term in the diagnostic field; thus, owing to its frequency, it becomes clinically and pathologically useless. Practically, perhaps simpler scoring protocols will increase the reproducibility [23].

In conclusion, 5% of the patients in our study presented with poorly differentiated OSCC at diagnosis. As previously stated, grade 3 tumors have worse prognosis and are more aggressive than grades 1 and 2 tumors; alternatively, grade 3 tumors are more sensitive to radiotherapy than other grade tumors [24–27]. The ability to predict which primary lesions are capable of early metastasis or poor prognosis would enable more individualized and aggressive therapy to be delivered to patients at higher risk of locoregional disease recurrence and death. In the future, we should focus on how to modify individual therapy for poorly differentiated OSCC to achieve improved outcomes.

## Supporting information

**S1 File.**
(PDF)

## Acknowledgments

The authors would like to thank Enago (www.enago.com) for the English language review.

## Author Contributions

**Conceptualization:** Nan-Chin Lin.

**Data curation:** Nan-Chin Lin.

**Formal analysis:** Nan-Chin Lin.

**Investigation:** Kuo-Yang Tsai.

**Methodology:** Jui-Ting Hsu.

**Supervision:** Jui-Ting Hsu, Kuo-Yang Tsai.

**Validation:** Jui-Ting Hsu.

**Visualization:** Jui-Ting Hsu.

**Writing – original draft:** Nan-Chin Lin.

**Writing – review & editing:** Jui-Ting Hsu, Kuo-Yang Tsai.

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
