## [Decision Letter · Decision Letter 0]

16 Jun 2020

PONE-D-20-10673

Survival and Clinicopathological Characteristics of Different Histological Grades of Oral Cavity Squamous Cell Carcinoma: A Single-Center Retrospective Study

PLOS ONE

Dear Dr. Tsai,

Thank you for submitting your manuscript to PLOS ONE. After careful consideration, the reviewers and i feel that it has merit but does not fully meet PLOS ONE’s publication criteria as it currently stands. Therefore, we invite you to submit a revised version of the manuscript that addresses the points raised during the review process.

We look forward to receiving your revised manuscript.

Kind regards,

Dipak Sapkota, PhD

Academic Editor

PLOS ONE

Journal Requirements:

2. In the ethics statement in the manuscript and in the online submission form, please provide additional information about the patient records used in your retrospective study, including: a) whether all data were fully anonymized before you accessed them; b) the date range (month and year) during which patients' medical records were accessed; and c) the source of the medical records analyzed in this work (e.g. hospital, institution or medical center name). If patients provided informed written consent to have data from their medical records used in research, please include this information.

 [The funders had no role in study design, data collection and analysis, decision to publish, or preparation of the manuscript.].

Reviewers' comments:

Reviewer's Responses to Questions

**Comments to the Author**

1. Is the manuscript technically sound, and do the data support the conclusions?

Reviewer #1: Yes

Reviewer #2: Partly

Reviewer #3: Yes

2. Has the statistical analysis been performed appropriately and rigorously? 

Reviewer #1: Yes

Reviewer #2: Yes

Reviewer #3: Yes

3. Have the authors made all data underlying the findings in their manuscript fully available?

Reviewer #1: Yes

Reviewer #2: Yes

Reviewer #3: No

4. Is the manuscript presented in an intelligible fashion and written in standard English?

Reviewer #1: Yes

Reviewer #2: No

Reviewer #3: No

5. Review Comments to the Author

Reviewer #1: The authors present a complete and well organised manuscript that highlights the importance of providing histological grade of oral squamous carcinomas. Although the significance of the findings might be deeply related to the demographical and social characteristics of the cohort (betel chewing for example), the main suggestions about the value of the histological grading in OSCC in this report are still universally valuable and in line with practice with other type of malignancies affecting other organs/systems and that are stratified by histological grade.

The authors exclude from the cohort cases of carcinomas of the lip not affecting labial mucosa, but Table 5 reports tumors of the lips, which is confusing. Indicating this group as "labial mucosa" sounds more correct and takes away this confusing.

The authors correctly point that the histological grade might be to a certain degree subjective, and here the grading is performed by only one individual, which should be pointed also in the discussion as one of the challenges with this manuscript.

Finally, the authors use OSCC but also SCC and should keep one form; in page 2 and page 15 "should not be disregarded" might be better expressed as for "should be followed more closely than OSCC in early stages but higher differentiation degree"; in page 12 it should be "poor differentiation" not "poor differentiate"; in page 15 I guess authors mean "diagnostic field" and not "diagnostic filed".

Reviewer #2: Dear authors,

Thank you for the opportunity to review this manuscript in a very interesting topic. I have several concerns about your manuscript pointed bellow. I strongly recommend an English language review. You should considerer to reestructure the manuscript, this atual version is presented in a very confuse way.

Abstract

- The abstract section is very confuse and do not reflect clearly the content of the study.

Introduction

- The introduction section is a little poor, the authors should consider to include one or two more paragraphs to state the problem to the readers.

Patients and Methods

- The authors have to add the number and the date of the ethical board protocol aprouvement. (page 3)

- In my point of view include lip squamous cell carcinoma is a bias in your study. It is well known that, in this location, the neoplasia have a complete different biological behaviour. (page 4)

- The authors have to explain better the histopathological classification, especially how it has been performed, this is the central part of the study. Only one pathologist performed the analysis? How many slides per day? Directly in a microscope? With an image manager program? (page 4)

- Has the immunohistochemistry been performed in a regular basis? How many cases? In wich conditions?(page 5)

- As mentioned, the inclusion of the lip is a bias in your study. In my point of view, the authors should exclude this region. (page 5)

Results

- In general the results were presented in a very confuse way, the authors should rewrite this part of the manuscript. The tables are very difficult to understant and some of them are unnecessary.

Discussion

- In my point of view the discussion section is a bit confuse and include a restrict number of references, considering this is a classic topic (only 18 references in total).

- The authors should rewrite the discussion section according to the reestructured results section in a possible future revised version.

Reviewer #3: Although the finding are significant, the manuscript is not written well. Therefore I advise you to get the help from a language expert. There is a significant amount of literature missing in the reference list.

---

## [Author Response · Author response to Decision Letter 0]

25 Jun 2020

General response:

 We sincerely thank the editor and all reviewers for their valuable feedback that we have used to improve the quality of our manuscript. The reviewer comments are laid out below in italicized font and specific concerns have been numbered. Our response is given in normal font and changes/additions to the manuscript are given in ”Track changes”.

 We have revised the manuscript extensively based on the reviewer’s comments. If there are any other modifications we could make, we would like very much to do so and we greatly appreciate your help. We hope that our manuscript will be considered for publication in your journal. Thank you very much.

Sincerely,

Kuo-Yang Tsai

Department of Oral and Maxillofacial Surgery, Changhua Christian Hospital

No. 235, Xuguang Rd, Changhua City, Changhua County 500, Taiwan

Phone: +886 933127916

Email: 72837@cch.org.tw

---

## [Decision Letter · Decision Letter 1]

21 Jul 2020

PONE-D-20-10673R1

Survival and Clinicopathological Characteristics of Different Histological Grades of Oral Cavity Squamous Cell Carcinoma: A Single-Center Retrospective Study

PLOS ONE

Dear Dr. Tsai,

Thank you for submitting your manuscript to PLOS ONE. After careful consideration, we still feel that the reviewers' comments are not satisfactorily addressed in the current version. Therefore, we invite you to submit a revised version of the manuscript that addresses the points raised during the review process.

We look forward to receiving your revised manuscript.

Kind regards,

Dipak Sapkota, PhD

Academic Editor

PLOS ONE

Reviewers' comments:

Reviewer's Responses to Questions

**Comments to the Author**

1. If the authors have adequately addressed your comments raised in a previous round of review and you feel that this manuscript is now acceptable for publication, you may indicate that here to bypass the “Comments to the Author” section, enter your conflict of interest statement in the “Confidential to Editor” section, and submit your "Accept" recommendation.

Reviewer #1: (No Response)

Reviewer #2: (No Response)

2. Is the manuscript technically sound, and do the data support the conclusions?

Reviewer #1: Partly

Reviewer #2: Partly

3. Has the statistical analysis been performed appropriately and rigorously? 

Reviewer #1: Yes

Reviewer #2: Yes

4. Have the authors made all data underlying the findings in their manuscript fully available?

Reviewer #1: Yes

Reviewer #2: Yes

5. Is the manuscript presented in an intelligible fashion and written in standard English?

Reviewer #1: Yes

Reviewer #2: No

6. Review Comments to the Author

Reviewer #1: I commend the authors for having worked with the text and accepting our suggestions, and the presentation of the data and the discussion are more complete and precise. The manuscript has improved on the language side as well, but there are still mistakes in the text (page 8 "...grade 2, years, and grade 3 OSCC..."; page 9 "...most..." instead of "...mostly..."; page 16 "re-wide excision" never heard before, I guess the authors mean "wider re-excision").

In page 17 the authors write that "...grading attempts are inexact, subjective, not reproducible." It is also my experience that most squamous carcinomas fall in the definition of moderate grade of differentiation, but the authors are quite sharp in their definition. And just using keratinising vs non keratinising could give the same problem and be potentially subjective. This has been dealt with by other authors who suggested a to-tier grading to improve the concordance of the grading [I can quickly suggest for example Krishnan, Karpagalvi...Patil, J Oral Maxillofac Pathology (2016) and Steigen, Søland...Olsen, J Oral Pathol Med (2020)].

Also, in the text the authors now write that two pathologists classified the material and not just one, but the numbers of the groups were the same as far as I can see, and the authors write nothing on how the two pathologists came to an agreement. Did the pathologists agree on all cases? In this case why do the authors write the grading is not reproducible?

This point (only one pathologist) and a point previously addressed (grade 3 cases should not be "disregarded") have not been addressed in the abstract, that should be corrected as well.

The text is written in good English all in all but the imprecisions deter from the scientific value of the study, real piety since I find the study really useful also in clinical settings, reminding pathologists to provide clinicians with a histological variable that can be evaluated on routinely provided HE slides.

Reviewer #2: The main questions addressed in the first round of revision were not properly answered. Including the lip OSCC is one of the major concern in the manuscript.

7. PLOS authors have the option to publish the peer review history of their article (what does this mean?). If published, this will include your full peer review and any attached files.

Reviewer #1: No

Reviewer #2: No

---

## [Author Response · Author response to Decision Letter 1]

24 Jul 2020

General response:

We sincerely thank the editor and all reviewers for their valuable feedback that we have used to improve the quality of our manuscript. The reviewer comments are laid out below in italicized font and specific concerns have been numbered. Our response is given in normal font and changes/additions to the manuscript are given in ”Track changes”.

Point-to-point response:

Reviewer #1: I commend the authors for having worked with the text and accepting our suggestions, and the presentation of the data and the discussion are more complete and precise. The manuscript has improved on the language side as well, but there are still mistakes in the text (page 8 "...grade 2, years, and grade 3 OSCC..."; page 9 "...most..." instead of "...mostly..."; page 16 "re-wide excision" never heard before, I guess the authors mean "wider re-excision"). 

Response: Thank you very much for your positive comments and suggestions. I have revised the manuscript accordingly. 

In page 17 the authors write that "...grading attempts are inexact, subjective, not reproducible." It is also my experience that most squamous carcinomas fall in the definition of moderate grade of differentiation, but the authors are quite sharp in their definition. And just using keratinising vs non keratinising could give the same problem and be potentially subjective. This has been dealt with by other authors who suggested a to-tier grading to improve the concordance of the grading [I can quickly suggest for example Krishnan, Karpagalvi...Patil, J Oral Maxillofac Pathology (2016) and Steigen, Søland...Olsen, J Oral Pathol Med (2020)].

Response: Thank you very much for your suggestion. I found the conclusion in Steigen et al. was similar to my point of view, they suggested that simpler/uncomplicated scoring protocols will increase the reproducibility. In their study, Intrarater comparison of WHO grading system, after in binary categorization, mean agreement could reach 92.5%. I have added the article into my reference and revised the manuscript in smooth tone. Thanks again for your valuable help. 

Also, in the text the authors now write that two pathologists classified the material and not just one, but the numbers of the groups were the same as far as I can see, and the authors write nothing on how the two pathologists came to an agreement. Did the pathologists agree on all cases? In this case why do the authors write the grading is not reproducible?

Response: In the present study, we investigated the grade of OSCC in retrospective fashion (based on final pathological report). The pathological report was accomplished by two pathologists within 10 days after surgery had been done. And the intervention of third pathologist was engaged if the first two pathologists could not reach a consensus. In our study we only presented the final agreement of the grade of tumors documented on the pathological report. Grading was achieved by human through microscope and it is inevitable that human made different judgment in those borderline cases. Steigen et al. also reported that simpler/uncomplicated scoring protocols will increase the reproducibility (two options better than three options, and three options better than fine or four options). In our study, we did not verify the reproducibility of OSCC grading, however, it is very hard to achieve 100% the same result when all our cases graded in the other center or other country. 

This point (only one pathologist) and a point previously addressed (grade 3 cases should not be "disregarded") have not been addressed in the abstract, that should be corrected as well. The text is written in good English all in all but the imprecisions deter from the scientific value of the study, real piety since I find the study really useful also in clinical settings, reminding pathologists to provide clinicians with a histological variable that can be evaluated on routinely provided HE slides. 

Response:

Thank you very much for your suggestion. I have revised the Abstract accordingly. In the present study, I would like to suggest clinicians do not overlook the importance of the grade of OSCC, in 2018 AJCC Head and Neck guideline, tumor invade depth was added in TNM staging. In somedays, histological grade of OSCC or a new grading system maybe added into TNM stage. 

 

Reviewer #2: The main questions addressed in the first round of revision were not properly answered. Including the lip OSCC is one of the major concern in the manuscript. 

Response: Thanks for your comments. In AJCC 2018 head and neck cancer guideline, oral cavity had included the mucosal lip. In Southeast Asia, the main factors contributing to mucosal lip SCC are betel nuts chewing, smoking and alcohol consumption and these factors are the same in the oral cavity SCC. Meanwhile, I have reviewed many researches for OSCC, and found these article also included mucosal lip SCC into their study (I take two study for example: Brinkman D, Callanan D, O'Shea R. Impact of 3 mm margin on risk of recurrence and survival in oral cancer. Oral Oncol. 2020;110:104883. doi:10.1016/j.oraloncology.2020.104883 and Mochizuki Y, Harada H, Ikuta M, et al. Clinical characteristics of multiple primary carcinomas of the oral cavity. Oral Oncol. 2015;51(2):182-189. doi:10.1016/j.oraloncology.2014.11.013 ). If you insist excluding the entity of the mucosal lip in the present study, we would like very much to do so and we greatly appreciate your help.

In the first round of revision, we have revised the manuscript extensively based on your suggestions. In the present study, we investigated the grade of OSCC in retrospective fashion (based on final pathological report). The pathological report was accomplished by two pathologists within 10 days after surgery had been done. And the intervention of third pathologist was engaged if the first two pathologists could not reach a consensus. Immunohistochemical confirmation for cytokeratin was performed to detect poorly differentiated OSCC because features of squamous differentiation were minimal or absent. In our study we only presented the final agreement of the grade of tumors documented on the pathological report. In the present study, I would like to suggest clinicians do not overlook the importance of the grade of OSCC, in 2018 AJCC Head and Neck guideline, tumor invade depth was added in TNM staging. In somedays, histological grade of OSCC or a new grading system maybe added into TNM stage. If there are any other modifications we could make, we would like very much to do so. 

We have revised the manuscript extensively based on the reviewer’s comments. If there are any other modifications we could make, we would like very much to do so and we greatly appreciate your help. We hope that our manuscript will be considered for publication in your journal. Thank you very much.

Sincerely,

Kuo-Yang Tsai

Department of Oral and Maxillofacial Surgery, Changhua Christian Hospital

No. 235, Xuguang Rd, Changhua City, Changhua County 500, Taiwan

Phone: +886 933127916

Email: 72837@cch.org.tw

---

## [Editor Report · Decision Letter 2]

30 Jul 2020

PONE-D-20-10673R2

Survival and clinicopathological characteristics of different histological grades of oral cavity squamous cell carcinoma: a single-center retrospective study

PLOS ONE

Dear Dr. Tsai,

Thank you for revising the manuscript according to the reviewers suggestions. The manuscript has improved significantly, but i feel that the manuscript will immensely benefit if you could improve the language (especially the consistent use of 'tense' in the method, result and the discussion sections) of your manuscript. Therefore, i invite you to submit a revised version of the manuscript. 

We look forward to receiving your revised manuscript.

Kind regards,

Dipak Sapkota, PhD

Academic Editor

PLOS ONE

Editor Comments:

Thank you for revising the manuscript according to the reviewers suggestions. The manuscript has improved significantly, but the manuscript still appears inconsistent with respect to the use of 'tense' in the method, result and the discussion sections. Please use a consistent tense to describe the current results as well as results from other researchers.

---

## [Author Response · Author response to Decision Letter 2]

4 Aug 2020

General response:

We sincerely thank the editor and all reviewers for their valuable feedback that we have used to improve the quality of our manuscript. The editor comments are laid out below in italicized font and specific concerns have been numbered. Our response is given in normal font and changes/additions to the manuscript are given in ”Track changes”.

Point-to-point response:

Editor: Thank you for revising the manuscript according to the reviewers suggestions. The manuscript has improved significantly, but the manuscript still appears inconsistent with respect to the use of 'tense' in the method, result and the discussion sections. Please use a consistent tense to describe the current results as well as results from other researchers.

Response: Thank you very much for your positive comments and suggestions. I have revised the manuscript accordingly. The changes to the manuscript are given in “Track changes”.

We have revised the manuscript extensively based on the yours comments. If there are any other modifications we could make, we would like very much to do so and we greatly appreciate your help. We hope that our manuscript will be considered for publication in your journal. Thank you very much.

Sincerely,

Kuo-Yang Tsai

Department of Oral and Maxillofacial Surgery, Changhua Christian Hospital

No. 235, Xuguang Rd, Changhua City, Changhua County 500, Taiwan

Phone: +886 933127916

Email: 72837@cch.org.tw

---

## [Editor Report · Decision Letter 3]

11 Aug 2020

Survival and clinicopathological characteristics of different histological grades of oral cavity squamous cell carcinoma: a single-center retrospective study

PONE-D-20-10673R3

Dear Dr. Tsai,

We’re pleased to inform you that your manuscript has been judged scientifically suitable for publication and will be formally accepted for publication once it meets all outstanding technical requirements.

Kind regards,

Dipak Sapkota, PhD

Academic Editor

PLOS ONE
---

## [Editor Report · Acceptance letter]

17 Aug 2020

PONE-D-20-10673R3 

Survival and clinicopathological characteristics of different histological grades of oral cavity squamous cell carcinoma: a single-center retrospective study 

Dear Dr. Tsai:

I'm pleased to inform you that your manuscript has been deemed suitable for publication in PLOS ONE. Congratulations! Your manuscript is now with our production department. 

Kind regards, 

on behalf of

Dr. Dipak Sapkota 

Academic Editor

PLOS ONE